# Petrographic Insights into the Evolution of Nano-Scale Organic Matter Pores with Organic Matter Conversion

Lei Zhou [1,2,3], Xingqiang Feng [1,2,3,*], Linyan Zhang [1,2,3], Lin Wu [1,2] and Rui Zhang [4]

1 Institute of Geomechanics, Chinese Academy of Geological Sciences, Beijing 100081, China; zhoulei4010@126.com (L.Z.)
2 Key Laboratory of Petroleum Geomechanics, China Geological Survey, Beijing 100081, China
3 Key Laboratory of Paleomagnetism and Tectonic Reconstruction, Ministry of Natural Resources, Beijing 100081, China
4 Well Testing Company, Qinghai Oilfield, Mangya 816499, China
* Correspondence: fxingqiang@mail.cgs.gov.cn; Tel.: +86-136-9303-4622

**Abstract:** To investigate the influence of organic matter conversion on the evolution of organic matter pores, fractional conversion ($TR_{HI}$) and loss of TOC ($TOC_L$) from the organic matter conversion of Middle Jurassic Dameigou Formation shale samples were calculated using petrographic analysis. The $TR_{HI}$ of organic matter varies from 0.30 to 0.88 and $TOC_L$ content ranges from 0.62% and 4.09%. Relative to samples of Type III organic matter in shales, type II samples exhibit higher $TR_{HI}$ and $TOC_L$ values. Petrographic calculations of $TR_{HI}$ reveal that the fractional conversion of different kerogens differs for the same thermal maturity level. The specific surface area ($S_{BET}$) ranges between 1.25 and 6.63 m$^2$/g and micropore surface area ($S_{mic}$) ranges between 4.16 and 21.27 m$^2$/g. Correlations between pore structure parameters and $TOC_L$ content are higher than those between pore structure parameters and TOC content. The original TOC content decreases with increasing maturity level owing to hydrocarbon generation from organic matter conversion. The development of organic matter pores depends mainly on organic matter conversion, which is influenced by the richness, organic maceral compositions, and thermal maturity of the organic matter. The contents of kaolinite, illite, and mixed-layer illite/smectite (I/S) in the studied shales are 17.83%–37.05%, 5.36%–11.31%, and 5.27%–14.36%, respectively. Pore structure parameters ($S_{BET}$ and $S_{mic}$) exhibit moderate positive correlations with illite content and I/S content, and moderate negative correlations with kaolinite content, indicating that different clay minerals have differential effects on pore structure.

**Keywords:** fractional conversion; organic matter; pore structure; shale





## 1. Introduction

Loucks et al. used the focused ion beam–scanning electron microscope (FIB–SEM) method to observe nanoscale pores within or between organic matter and reported the morphology, distribution, and genesis of the pores [1–5]. Various measurement techniques, including USANS/SANS [6–8], X-ray micro-computed tomography [9,10], subcritical gas adsorption [11], and nuclear magnetic resonance [12,13], have been used to characterize the microstructural features of organic matter pores. The occurrence and evolution of organic matter pores are correlated with organic richness, thermal maturity, kerogen type, and maceral composition [14–17].

Organic matter pores contribute more to the micropore surface area, which provides more sorption sites for methane [3]. Due to the conversion of organic matter to hydrocarbon, organic matter pores increase with increasing thermal maturity [4,5]. The proportion of organic matter pores in pore network systems could increase to 40%–80% [2,5]. Although organic matter pores contribute less to mesopores and macropores, leading to an increase in porosity, organic matter pores lead to more interconnections between micropores and meso/macropores, resulting in a higher permeability [2]. Organic matter pores within a

size scale of several to dozens of nanometers are a critical indicator for evaluating shale gas reservoirs and methane adsorption capacity [1–3].

Curtis et al. and Yang et al. analyzed the pore structure features of shales with different maturity levels and found that pore surface area and pore volume exhibited positive correlations with maturity [17,18]. Investigation of both organic matter type and maceral composition is required to obtain a more comprehensive understanding of the evolution of organic matter pores. Using the subcritical gas adsorption method, Wei et al. investigated the pore structure of shales ranging in maturity from immature to post-mature before and after the extraction of soxhlet-extractable bitumen and oil and found that maturity had an important influence on the evolution of organic pores [3]. Recent studies have used anhydrous and hydrous pyrolysis experiments to understand how organic matter pores evolve with increasing thermal maturity [19–22]. The conversion of organic matter during thermal maturation results in an increase in micro/mesopores. The evolution of organic matter pores occurs in three stages: formation, development, and destruction [4,23]. Sun et al. used hydrous pyrolysis to investigate the evolution of pore structure and found that pore volume increases with a greater loss of total organic carbon ($TOC_L$) caused by organic matter conversion [24].

In this study, the original TOC content ($TOC_o$), $TOC_L$, and the conversion ratio of organic matter to hydrocarbons ($TR_{HI}$) were reconstructed through petrographic analysis. Micropore and mesopore structural parameters were evaluated by low-pressure $N_2$ and $CO_2$ adsorption methods, respectively. The results show that a comparison of pore structures at different stages of organic matter conversion (i.e., different $TOC_L$ contents), using petrographic analysis, provides information on the formation and development of organic matter pores during thermal evolution.

## 2. Samples and Methods

### 2.1. Samples

Although different macerals have different physicochemical properties depending on maturity, the optical features of different macerals could evolve to be consistent [14]. It is difficult to distinguish vitrinites from inertinites, liptinites, and solid bitumen at a very mature stage. Due to low maturity, the Middle Jurassic Dameigou Formation was chosen for this study. The Dameigou Formation (DMG) is a Middle Jurassic sedimentary unit in the northern margin of the Qaidam basin in China (Figure 1). This formation has been explored for oil and gas since the 1980s [25]. It consists of open-lacustrine, marginal–lacustrine, and fluvial deposits that reflect a humid climate during the Early to Middle Jurassic [26–28]. In this study, 10 shale samples were collected from a DMG outcrop section (Figure 2).

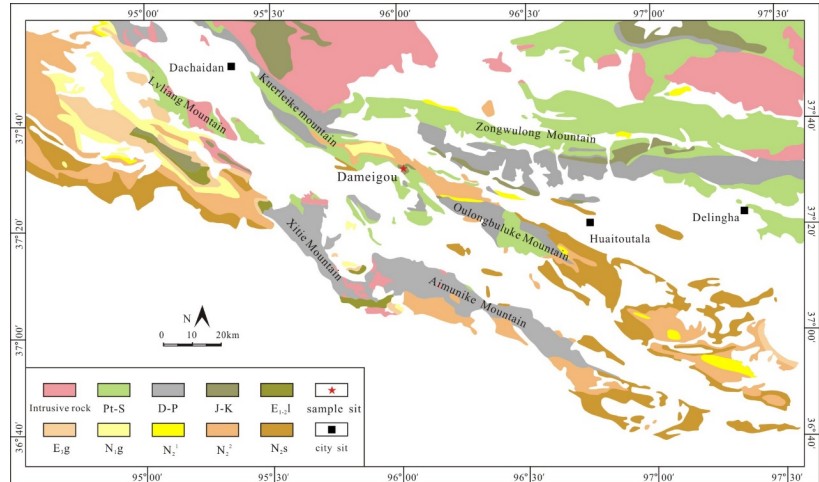

**Figure 1.** Simplified geological map of the Northern Qaidam Basin and location of the sampled Dameigou section.

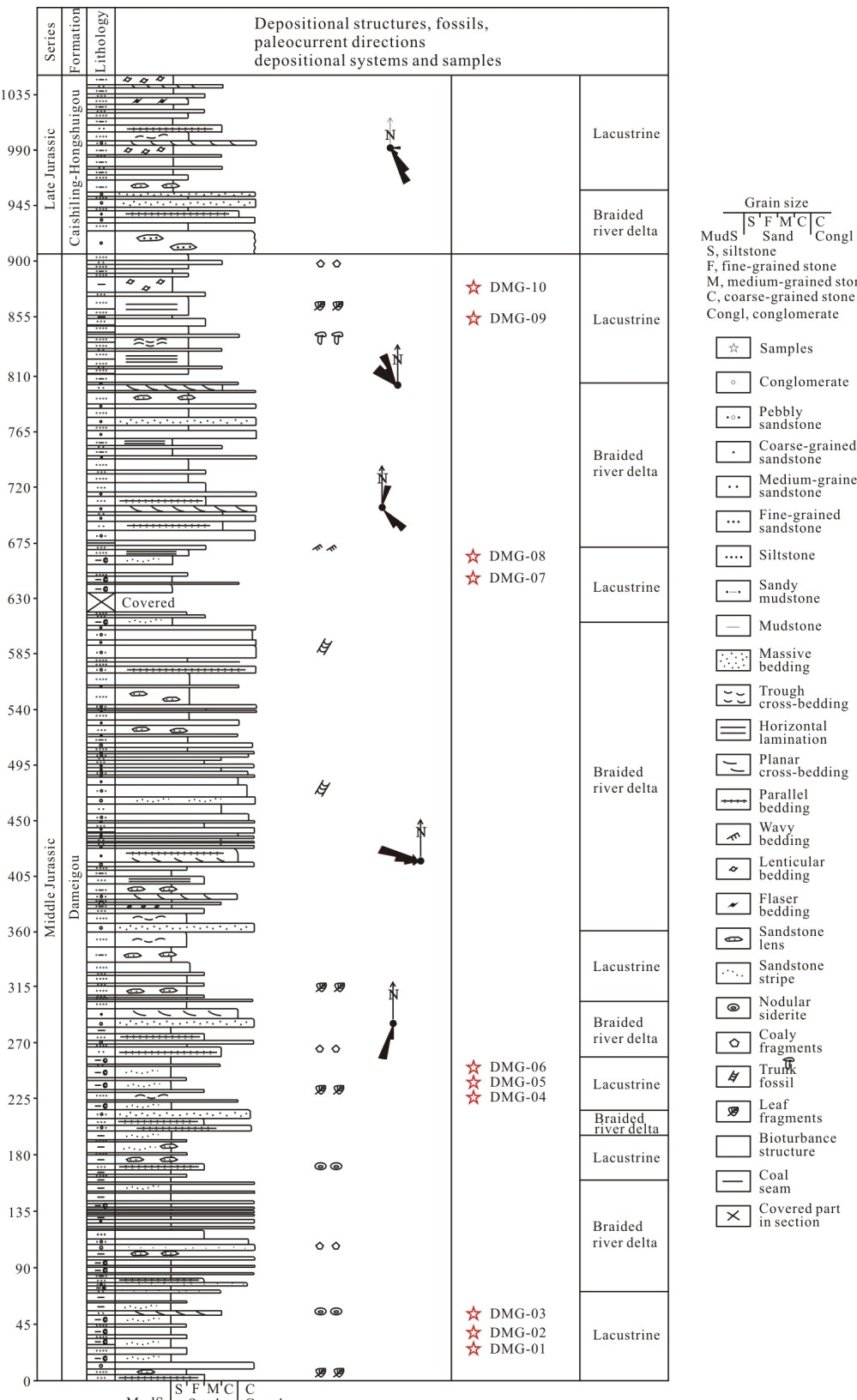

**Figure 2.** Stratigraphic column for the Lower–Upper Jurassic of the northern Qaidam Basin showing the locations of the studied samples of Dameigou Formation shale.

### 2.2. Geochemical and Mineralogical Analyses

The 10 samples were ground to <80 mesh. Each powder sample, weighing 500 mg, was initially treated using hydrochloric acid until the absence of bubbles indicated the complete removal of carbonates. The samples were then washed in distilled water. The TOC content was subsequently analyzed using a Leco CS 244 carbon–sulfur analyzer (National Research Center for Geoanalysis, Chinese Academy of Geological Sciences, Beijing, China). The Rock-Eval 6 Turbo analyzer (National Research Center for Geoanalysis, Chinese Academy of Geological Sciences, Beijing, China) was set at an initial temperature of 300 °C, which was increased to 600 °C at a rate of 25 °C/min. The free hydrocarbons ($S_1$), pyrolyzable hydrocarbons ($S_2$), hydrogen index (*HI*), and production index (*PI*) were obtained from the Rock-Eval pyrolysis. Vitrinite reflectance ($R_o$), a well-established indicator of thermal maturity, was measured using an MPV-SP microscope equipped with an oil immersion lens. Maceral compositions and vitrinite reflectance were determined by making at least 50 measurements. Mineral compositions were measured using a TD-3500 X-ray diffractometer (TD-3500, Dandong Tongda Technology Co., LTD, Dandong, Liaoning, China), calculated by semi-quantification of the area within major peaks. TD-3500 X-ray diffractometer is operating at 30 kV and 20 mA with a Cu K$\alpha$ radiation. The intensity data were collected in the 2θ range of 3–85° at a rate of 4°/min.

### 2.3. Low-Pressure $N_2$ and $CO_2$ Adsorption Analyses and FE–SEM Imaging

The 10 samples, each weighing 3–4 g, were degassed for 11 h at 110 °C under vacuum prior to adsorption analysis. Low-pressure $N_2$ adsorption analysis was conducted using a Quadrasorb$^{TM}$ SI Surface Area Analyzer (Quadrasorb SI, Quantachrome Instruments, Boynton Beach, FL, USA) at 77 K. The gas adsorption procedure and parameters followed those described by Tian et al. [29]. The specific surface area and pore size distribution of the samples were estimated by reference to adsorption isotherms using the Brunauer-Emmett-Teller (BET) [30] and Barrett-Joyner-Halenda (BJH) [31] methods. The micropore surface area and volume of the samples were estimated using the Dubinin–Astakhov (D–A) model, which was applied to $CO_2$ adsorption analysis conducted at 273.15 K. Thin sections, which were polished using Ar ions, were observed using a Quanta 450 field emission-scanning electron microscope (FE-SEM) (Quanta 450 FEG, FEI Company, Hillsboro, OR, USA).

### 2.4. Petrographic Analysis

The original hydrogen index ($HI_o$) was calculated using the volume percentages of alginate (*A*), liptinite (*L*), vitrinite (*V*), and inertinite (*I*) as follows [32]:

$$HI_o = \frac{A^P \times 750}{100} + \frac{L^P \times 450}{100} + \frac{V^P \times 125}{100} + \frac{I^P \times 50}{100} \tag{1}$$

The fractional conversion of organic matter ($TR_{HI}$) was calculated using *HI*, $HI_o$, *PI*, and the original production index ($PI_o$) as follows [14]:

$$TR_{HI} = 1 - \frac{HI \times [1200 - HI_o \times (1 - PI_o)]}{HI_o \times [1200 - HI \times (1 - PI)]} \tag{2}$$

where $PI_o$ was determined to be 0.02 [14]. The maximum *HI* is 1200, assuming that 83.33% carbons in hydrocarbons is generated.

$TOC_o$ was calculated using $HI_o$ and $TR_{HI}$ as follows [14]:

$$TOC_o = \frac{83.33 \times HI \times \left(\frac{TOC}{1+k}\right)}{HI_o \times (1 - TR_{HI}) \times (83.33 - \left(\frac{TOC}{1+k}\right)) - HI \times \left(\frac{TOC}{1+k}\right)} \tag{3}$$

where k is a correction factor, k = $TR_{HI} \times C_R$, with $C_R$ = 50% for type I kerogen, $C_R$ = 15% for type II kerogen, and $C_R$ = 0 for type III kerogen.

$TOC_L$ from organic matter conversion was calculated using $TOC_o$ and TOC as follows:

$$TOC_L = TOC_o - TOC \tag{4}$$

## 3. Results

### 3.1. Shale Mineralogical Composition

The DMG shale samples have clay mineral contents of 40.2%–55.3% and quartz contents of 29.5%–50.0% (Table 1). Compared with marine shales from China [13], the samples contain higher amounts of clay minerals and lower amounts of quartz. The vertical heterogeneity of clay minerals and quartz is due to deposition conditions, which respond to astronomically forced climate changes [33,34]. Moreover, the quartz content shows no clear relationship with TOC content, in contrast to marine shales. The DMG shales were deposited in open and marginal lacustrine settings [25–27], with the detritus transported from the Quanji Block by fluvial systems [28]. The difference in quartz contents between the DMG lacustrine shales and the Chinese marine shales suggests that most of the quartz in the studied lacustrine shales was derived from terrigenous detrital sources rather than biogenic sources. The siderite, plagioclase, and K-feldspar contents of the DMG shales are 2.3%–17.1%, 1.2%–6.1%, and 0%–4.9%, respectively.

**Table 1.** Results of XRD analysis and $N_2$ and $CO_2$ adsorption analysis of DMG shale samples.

| Sample ID | XRD Analysis | | | | | | | | $N_2$ Adsorption | | $CO_2$ Adsorption | |
|---|---|---|---|---|---|---|---|---|---|---|---|---|
| | Clay (%) | Qtz (%) | K-f (%) | Plag (%) | Sid (%) | I/S (%) | I (%) | Kaol (%) | $S_{BET}$ ($m^2$/g) | $V_p$ (ml/100 g) | $S_{mic}$ ($m^2$/g) | $V_{mic}$ (ml/g) |
| DMG-01 | 43.5 | 43.5 | 1.7 | 6.1 | 5.2 | 14.36 | 11.31 | 17.83 | 6.63 | 1.85 | 21.27 | 0.008 |
| DMG-02 | 50.7 | 39.6 | 1.8 | 5.1 | 2.8 | 10.64 | 9.13 | 30.93 | 6.42 | 1.78 | 15.99 | 0.006 |
| DMG-03 | 55.3 | 37.9 | | 2.6 | 4.2 | 8.3 | 9.95 | 37.05 | 4.59 | 1.48 | 13.41 | 0.005 |
| DMG-04 | 40.2 | 50 | 2 | 4.7 | 3.1 | 12.06 | 8.04 | 20.1 | 5.90 | 2.05 | 19.95 | 0.005 |
| DMG-05 | 41.2 | 46.9 | 3.4 | 1.9 | 6.6 | 7 | 5.36 | 28.84 | 1.69 | 0.96 | 4.90 | 0.002 |
| DMG-06 | 45.1 | 31.8 | 2.3 | 3.7 | 17.1 | 10.37 | 9.02 | 25.71 | 2.27 | 1.02 | 12.39 | 0.005 |
| DMG-07 | 52.4 | 41.8 | 2.3 | 1.2 | 2.3 | 7.34 | 7.33 | 37.73 | 3.40 | 1.81 | 4.16 | 0.002 |
| DMG-08 | 45.3 | 45.9 | 2.1 | 2.8 | 3.9 | 5.27 | 5.53 | 34.5 | 2.22 | 1.30 | 14.18 | 0.005 |
| DMG-09 | 52 | 29.5 | | 3.1 | 15.4 | 11.44 | 8.32 | 32.24 | 1.25 | 0.59 | 8.47 | 0.003 |
| DMG-10 | 48.9 | 40.3 | 4.9 | 3.4 | 2.5 | 10.75 | 7.34 | 30.81 | 4.44 | 1.83 | 9.85 | 0.004 |

Qtz: quartz, K-f: K-feldspar, Plag: plagioclase, Sid: Siderite, I/S: illite/smectitec mixed layer, I: illite, Kaol: kaolinite, $S_{BET}$: BET surface area from nitrogen adsorption, $V_p$: Single point adsorption total pore volume from nitrogen adsorption, $S_{mic}$: micropore surface area from $CO_2$ adsorption, $V_{mic}$: micropore volume from $CO_2$ adsorption.

Kaolinite contents of the DMG shales range from 17.83% to 37.05%, accounting for 41%–76% of the total clay mineral content (Table 1). Illite and mixed layer illite/smectite (I/S) contents range from 5.36% to 11.31% and 5.27% to 14.36%, respectively. Compared with over-mature Silurian Longmaxi Formation shales from China and shales from the United States [33,34], kaolinite, rather than illite and mixed-layer I/S, is dominant in the clay minerals of the DMG shales.

### 3.2. Pore Structure from $N_2$ Adsorption and $CO_2$ Adsorption

The low-pressure $N_2$ adsorption/desorption isotherms for the DMG shale samples are presented in Figure 3. The $N_2$ adsorption isotherms are type II with inflection point B, which signifies the completion of monolayer adsorption and the commencement of multilayer adsorption [35]. For $P/P_o$ values of >0.5, the $N_2$ adsorption/desorption isotherms of the shales exhibit hysteresis loops, attributable to capillary condensation within mesopores and macropores [36]. The hysteresis loops could be divided into two types: Type H2 and Type H3. In comparison with the low-pressure $N_2$ adsorption isotherms, the $CO_2$ adsorption isotherms for the studied shales closely resemble those of type I (Figure 4), suggesting the development of micropores in the DMG shales.

**Figure 3.** N$_2$ adsorption-desorption isotherms of DMG shale.

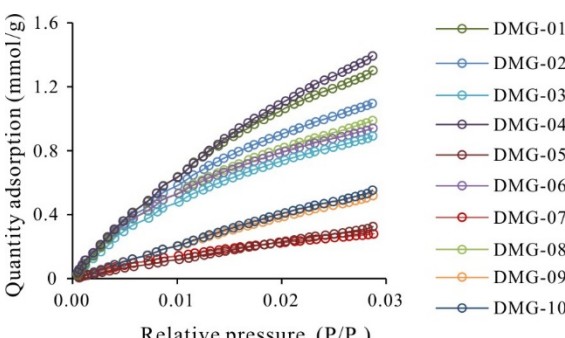

**Figure 4.** CO$_2$ adsorption isotherms of DMG shale.

Owing to the effects of tensile strength and the pore network on pore microstructure determined using N$_2$ desorption isotherms [37,38], the specific surface area ($S_{BET}$) and pore volume ($V_P$) were calculated for the DMG samples using N$_2$ adsorption isotherms through

the BET and BJH models, respectively. $S_{BET}$ ranges from 1.25 to 6.63 m$^2$/g, and $V_p$ from 0.59 to 2.05 mL/100 g (Table 1). The micropore surface area ($S_{mic}$) derived from the $CO_2$ adsorption isotherms ranges from 4.16 to 21.27 m$^2$/g, and the micropore volume ($V_{mic}$) derived from the $CO_2$ adsorption isotherms ranges from 0.002 to 0.008 mL/g.

### 3.3. Petrographic Analysis for Organic Matter Conversion

The TOC contents of the DMG shales are 2.74%–6.62% (Table 2). Values of $R_o$ for the DMG shales range from 0.82% to 0.83%, indicating the early to peak mature stage. $S_1$ and $S_2$ values of the samples vary between 0.09 and 0.85 mg/g and between 3.65 and 6.39 mg/g, respectively. Values of *HI* lie in the range of 55 to 171 mg HC/g TOC, and $T_{max}$ varies between 434 and 440 °C. Owing to the conversion of organic matter to hydrocarbons, *HI* values decrease with increasing maturity level, which can lead to the misleading identification of kerogen types if based solely on bivariate plots of HI and $T_{max}$ [32]. The type index of kerogen (TI), which is based on maceral composition [39], was also applied to identify kerogen types in this study. Two groups of samples are distinguished. The first group of five samples (DMG-05, DMG-06, DMG-08, DMG-09, and DMG-10) contain higher proportions of vitrinites and inertinites in the macerals (Table 3), and the TI values of these samples are <0, indicating a predominance of type III organic matter. The second group of five samples (DMG-01, DMG-02, DMG-03, DMG-04, and DMG-07), which contain higher proportions of alginites and liptinites, have TI values of 5–31, indicating a predominance of type II organic matter.

**Table 2.** Total organic Carbon content, vitrinite reflectance, and rock-eval pyrolysis for the DMG shale samples.

| Sample ID | TOC (%) | $R_o$ (%) | $S_1$ (mg/g) | $S_2$ (mg/g) | $T_{max}$ °C | HI (mg/g) | PI | $HI_o$ (mg/g) | $TR_{HI}$ | $TOC_o$ (%) | $TOC_L$ (%) |
|---|---|---|---|---|---|---|---|---|---|---|---|
| DMG-01 | 6.35 | 0.82 | 0.71 | 5.60 | 440 | 88 | 0.11 | 483 | 0.88 | 10.44 | 4.09 |
| DMG-02 | 5.26 | 0.83 | 0.32 | 4.79 | 436 | 91 | 0.06 | 406 | 0.84 | 7.50 | 2.24 |
| DMG-03 | 4.44 | 0.83 | 0.73 | 3.71 | 435 | 84 | 0.16 | 445 | 0.87 | 6.58 | 2.14 |
| DMG-04 | 5.00 | 0.83 | 0.28 | 3.97 | 435 | 79 | 0.07 | 439 | 0.88 | 7.43 | 2.43 |
| DMG-05 | 2.74 | 0.83 | 0.22 | 4.69 | 434 | 171 | 0.04 | 296 | 0.49 | 3.36 | 0.62 |
| DMG-06 | 3.79 | 0.83 | 0.09 | 5.13 | 438 | 135 | 0.02 | 289 | 0.60 | 4.89 | 1.10 |
| DMG-07 | 2.86 | 0.83 | 0.15 | 4.25 | 438 | 149 | 0.03 | 370 | 0.68 | 3.52 | 0.66 |
| DMG-08 | 6.62 | 0.83 | 0.21 | 3.65 | 436 | 55 | 0.05 | 145 | 0.65 | 8.61 | 1.99 |
| DMG-09 | 5.08 | 0.82 | 0.85 | 4.04 | 436 | 80 | 0.17 | 136 | 0.45 | 6.18 | 1.10 |
| DMG-10 | 5.68 | 0.82 | 0.59 | 6.39 | 438 | 113 | 0.08 | 154 | 0.30 | 6.90 | 1.22 |

**Table 3.** Maceral analysis and the type index (TI) of organic matter of the DMG shale samples.

| Sample ID | V | | | | | I | | | L | | | | A | | TI |
| | Te | CoD | Co | Vd | $V^p$/% | Sem | Ind | $I^p$/% | Spo | Cut | Re | Lip | $L^p$/% | Alg | $A^p$/% | |
|---|---|---|---|---|---|---|---|---|---|---|---|---|---|---|---|---|
| DMG-01 | 1.8 | 2.1 | | 2.3 | 30.85 | 0.8 | 0.7 | 7.46 | 0.6 | | | 0.9 | 7.46 | 10.9 | 55.23 | 27 |
| DMG-02 | 1.4 | 2.2 | | 1.6 | 24.53 | 0.8 | 0.3 | 5.19 | 7.2 | 0.2 | 0.7 | 2.8 | 51.42 | 4 | 18.87 | 21 |
| DMG-03 | | 1.3 | | 1.6 | 20.14 | 0.4 | 0.2 | 4.17 | 3.4 | 0.6 | 0.4 | 2.8 | 50.00 | 3.7 | 25.69 | 31 |
| DMG-04 | | 1.6 | | 2.1 | 24.18 | 0.4 | 0.1 | 3.27 | 5.4 | | 0.4 | 1.2 | 45.75 | 4.1 | 26.80 | 28 |
| DMG-05 | | 0.8 | | 1.2 | 51.28 | 0.2 | 0.2 | 10.26 | | | | 0.8 | 20.51 | 0.7 | 17.95 | −21 |
| DMG-06 | | 3.2 | | 0.4 | 51.43 | 0.3 | 0.2 | 7.14 | 1.2 | 0.5 | | 0.4 | 30.00 | 0.8 | 11.43 | −19 |
| DMG-07 | | 1.5 | | 1.4 | 36.25 | 0.6 | | 7.50 | 1.9 | | | 0.8 | 33.75 | 1.8 | 22.50 | 5 |
| DMG-08 | 3.4 | 28.8 | | 3.6 | 93.96 | | | | | | 0.6 | 1.7 | 6.04 | | | −67 |
| DMG-09 | 18.5 | 8.9 | 1.6 | 4.8 | 68.84 | 8.5 | 2.6 | 22.61 | 1.8 | | 0.8 | 1.6 | 8.55 | | | −70 |
| DMG-10 | 10.2 | 6.5 | | 3.6 | 84.94 | 0.7 | 0.5 | 5.02 | 1.6 | 0.3 | | 0.5 | 10.04 | | | −64 |

(1) V: vitrinite macerals, Te: telinite, Cod: collodetrinite, Co: collotelinite, Vd: vitrodetrinite, $V^p$: volume percentage of vitrinite, I: inertinite macerals, Sem: Semifusinite, Ind: inertodetrinite, $I^p$: volume percentage of inertinite, L: liptinite macerals, Spo: sporinite, Cut: cutinite, Re: resinite, Lip: liptodetrinite, $L^p$: volume percentage of liptinite, A: alginite macerals, Alg: alginite $A^p$: volume percentage of alginite; (2) $TI = 100 \times A^p + 50 \times L^p - 75 \times V^p - 100 \times I^p$; $TI \geq 80$, 80–40, 40–0 and <0 indicate type I, II$_1$, II$_2$, III, respectively.

The original hydrogen index ($HI_o$) reconstructed using petrographic analysis varies between 136 and 483 mg HC/g TOC. Values of $TR_{HI}$ of organic matter range from 0.30 to 0.88 (Table 2), indicating that most of the pyrolyzable kerogen in the DMG shale samples has been converted to hydrocarbons. A significant difference in $TR_{HI}$ is observed between type II samples ($TR_{HI}$ = 0.68–0.88) and type III ($TR_{HI}$ = 0.30–0.65). $TOC_o$ content is 3.36%–10.44%, and $TOC_L$ content calculated on the basis of organic matter conversion is 0.62%–4.09%. Type III samples have lower $TOC_L$ contents from organic matter conversion compared with type II samples (except for DMG-07). Sample DMG-07 consists predominantly of alginites and liptinites, but also contains vitrinites and inertinites (Table 3), leading to lower values of $TR_{HI}$ and $TOC_L$. In summary, relative to type III samples, type II samples exhibit a greater fractional conversion of organic matter for the generation of hydrocarbons and higher $TOC_L$ contents.

## 4. Discussion

### 4.1. Effect of Clay Minerals on Pore Structure

To investigate the influence of clay minerals on pore structure in the DMG shale, plots of total clay mineral content, illite content, I/S content, kaolinite content, and pore structure parameters are presented in Figure 5. Neither $S_{BET}$ nor $S_{mic}$ show a clear correlation with total clay mineral content (Figure 5a,e). $S_{BET}$ and $S_{mic}$ show moderate positive correlations with illite content and I/S content and moderate negative correlations with kaolinite content. The relationships of $S_{BET}$ and $S_{mic}$ with the contents of different clay minerals suggest that pore structure parameters are correlated with clay mineral type and that these correlations differ from one clay mineral type to another for a given clay content, leading to the lack of (or obscuring any) relationship between pore structure parameters and total clay mineral content.

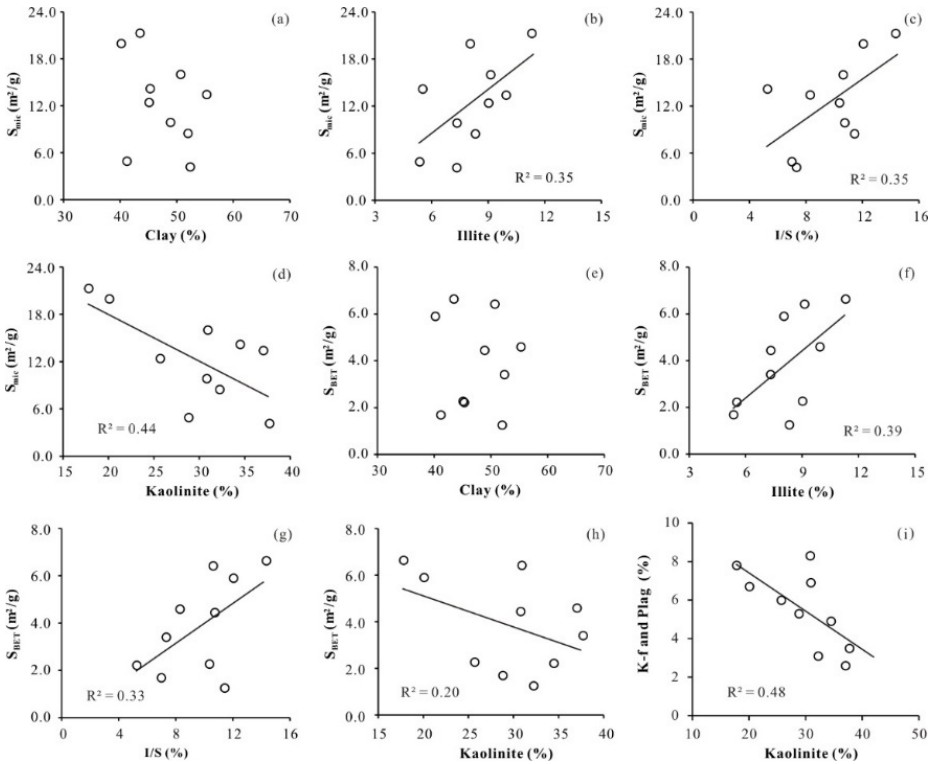

**Figure 5.** Relationships between pore structure parameters and clay minerals, and between kaolinite and K-feldspar and plagioclase. (**a**) Clay vs. $S_{mic}$, (**b**) Illite vs. $S_{mic}$, (**c**) I/S vs. $S_{mic}$, (**d**) Kaolinite vs. $S_{mic}$, (**e**) Clay vs. $S_{BET}$, (**f**) Illite vs. $S_{BET}$, (**g**) I/S vs. $S_{BET}$, (**h**) Kaolinite vs. $S_{BETc}$, (**i**) Kaolinite vs. K-feldspar and plagioclase.

Kaolinite shows a clear negative correlation with K-feldspar and plagioclase (Figure 5i). Kaolinite can originate from the dissolution of aluminosilicate minerals, such as detrital micas and feldspars, under moderately acidic conditions or can form in shales during diagenesis [40]. The morphology of kaolinite, characterized using an FE–SEM, resembles thick stacks or is vermicular (Figure 6). As the DMG shale is categorized as early to peak mature stage, the thermal degradation of organic matter is inferred to have led to hydrocarbon generation. This conversion of kerogen to hydrocarbons is accompanied by the generation of peripheral carboxylic acids, resulting in the dissolution and conversion of K-feldspar to kaolinite and authigenic quartz [41–43]. Owing to its standard 1:1 structure, which lacks internal surface area [44], kaolinite has a smaller $N_2$-specific surface area and $CO_2$ micropore surface area compared with illite and smectite [19]. The formation of kaolinite reduces the primary porosity and contributes to a decrease in secondary porosity [34], resulting in a weak negative correlation between kaolinite content and pore structure parameters.

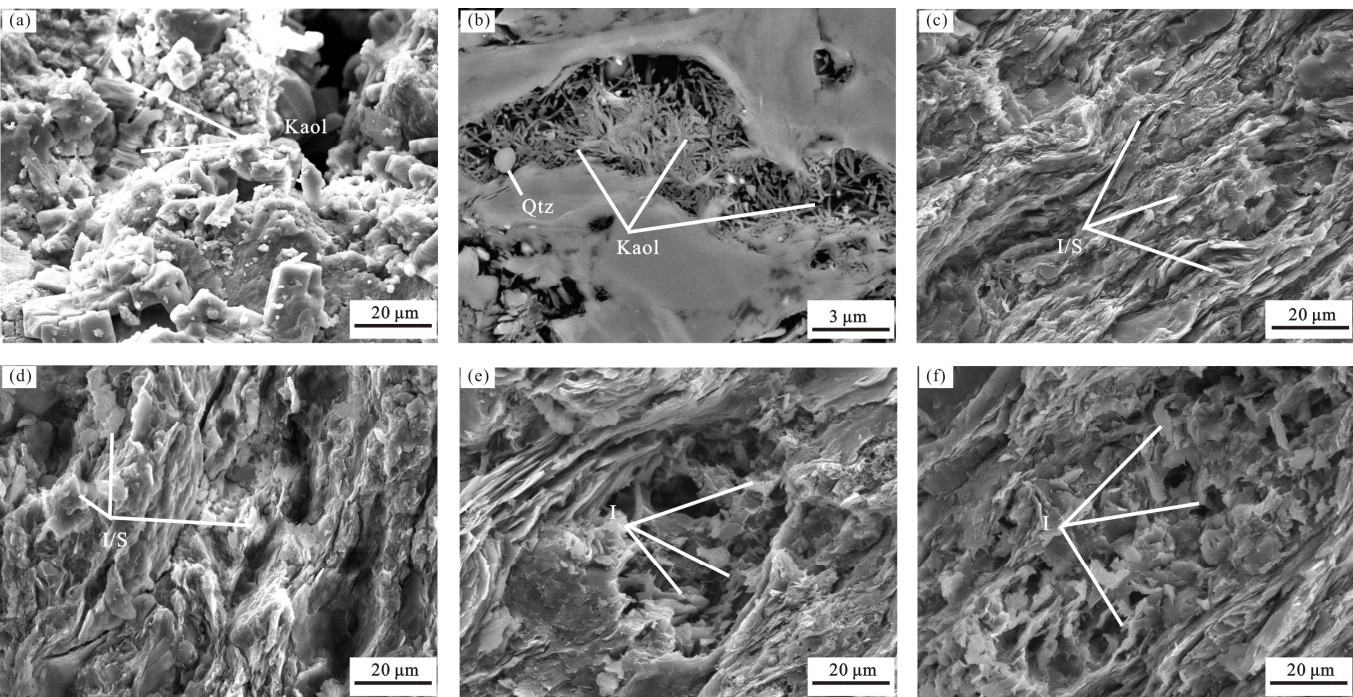

**Figure 6.** FE-SEM photomicrographs of clay minerals in DMG shale. Qtz: quartz, I/S: illite/smectitec mixed layer, I: illite, Kaol: kaolinite. (**a**) Kaolnite, (**b**) Kaolinite and quartz, (**c**) illite/smectitec mixed layer, (**d**) illite/smectitec mixed layer, (**e**) illite, (**f**) illite.

Generally, mixed-layer I/S and illite are the intermediate and final products of the dia-genetic conversion of smectite to illite, respectively [45]. The dissolution of feldspar, which is involved in kaolinite formation, is regarded as the main source of $K^+$ and $Al^{3+}$ [43,46] and promotes the transformation of smectite to illite and I/S. The DMG shales contain feldspar but not smectite, possibly indicating that all of the smectite has been transformed to illite or mixed-layer I/S. The illite and mixed-layer I/S in the DMG shale are ribbon-like and lath-shaped, respectively. In comparison with kaolinite, illite has a larger specific surface area and micropore surface area [16]. The release of interstitial water from smectite during illitization may lead to the development of overpressure reservoirs [47], which is beneficial for the preservation of primary porosity. The formation of illite and mixed-layer I/S increases the primary porosity and contributes to greater secondary porosity, resulting in the observed moderate positive correlations of pore structure parameters with illite content and I/S content.

### 4.2. Effect of Organic Matter Conversion on Pore Structure

A weak/moderate positive correlation is observed between $S_{mic}$ and TOC content ($R^2 = 0.48$) for the DMG shales, and a weak positive or no correlation ($R^2 = 0.15$) is observed between $S_{BET}$ and TOC content (Figure 7). These results are similar to the pore structure–TOC relationships for shales of the Lower Jurassic Gordondale Member from the Western Canadian Basin, together indicating that neither micropore surface area nor BET surface area have meaningful relationships with TOC content [16]. Values of vitrinite reflectance for the samples range from 0.82% to 0.83%, indicating that DMG shales fall within the oil window. Several studies have reported that vitrinite reflectance values of 0.5%–0.6% are indicative of the initiation of thermogenic gas generation. The $TR_{HI}$ values of the studied samples, which range between 0.30 and 0.88, indicate that organic matter has been converted into hydrocarbons. Owing to the conversion of kerogen to hydrocarbons, organic pores have developed in the DMG shale. Organic pores are discrete features and their degree of connectivity within the kerogen is poor, relative to organic pores in mature or over-mature shale (Figure 8). Hysteresis loops of DMG-01, DMG-02, DMG-03, and DMG-04 samples are type H2, while hysteresis loops of DMG-05, DMG-05, DMG-07, DMG-08, DMG-09, and DMG-10 samples are type H3. The type H2 loop represents inkbottle-shaped pores, and the type H3 loop is indicative of silt-shaped pores formed within aggregates of plate-like particles. Organic pores are ellipsoidal or spherical and the pore size of an organic pore is usually less than 20 nm. In addition, nanoscale intraparticle and interparticle pores are commonly found in clay minerals (Figure 8). DMG-01, DMG-02, DMG-03, and DMG-04 samples have more $TOC_L$ content, indicative of organic matter conversion and organic matter pore generation, relative to DMG-05, DMG-05, DMG-07, DMG-08, DMG-09, and DMG-10 samples. In addition, DMG-05, DMG-05, DMG-07, DMG-08, DMG-09, and DMG-10 samples have more clay mineral plate-like pores.

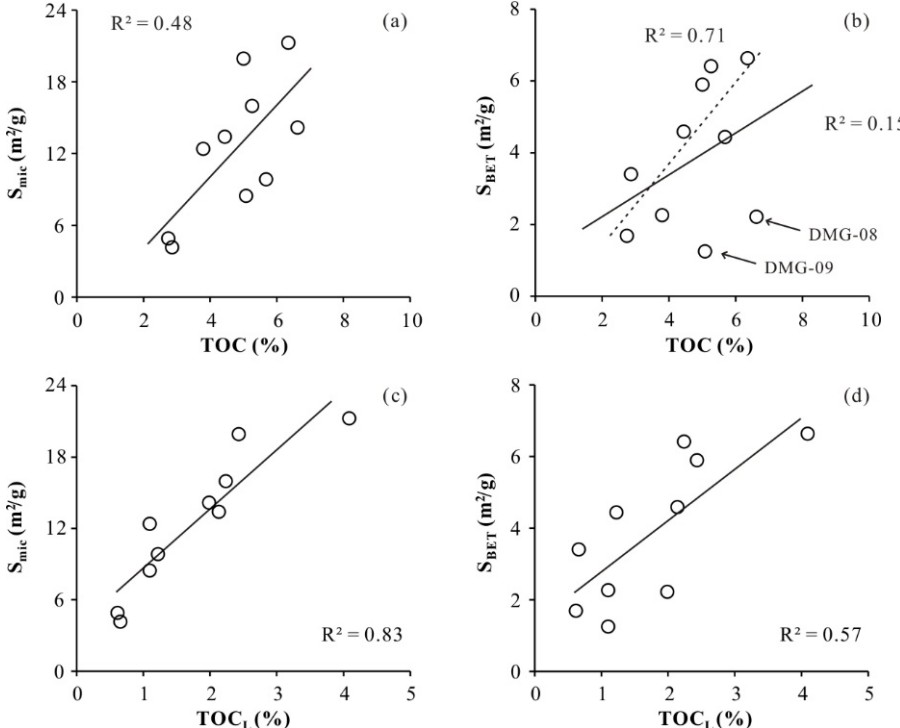

**Figure 7.** Relationships between pore structure parameters and organics. $S_{mic}$ vs. TOC (**a**); $S_{BET}$ vs. TOC (**b**); $S_{mic}$ vs. $TOC_L$ (**c**); $S_{BET}$ vs $TOC_L$ (**d**).

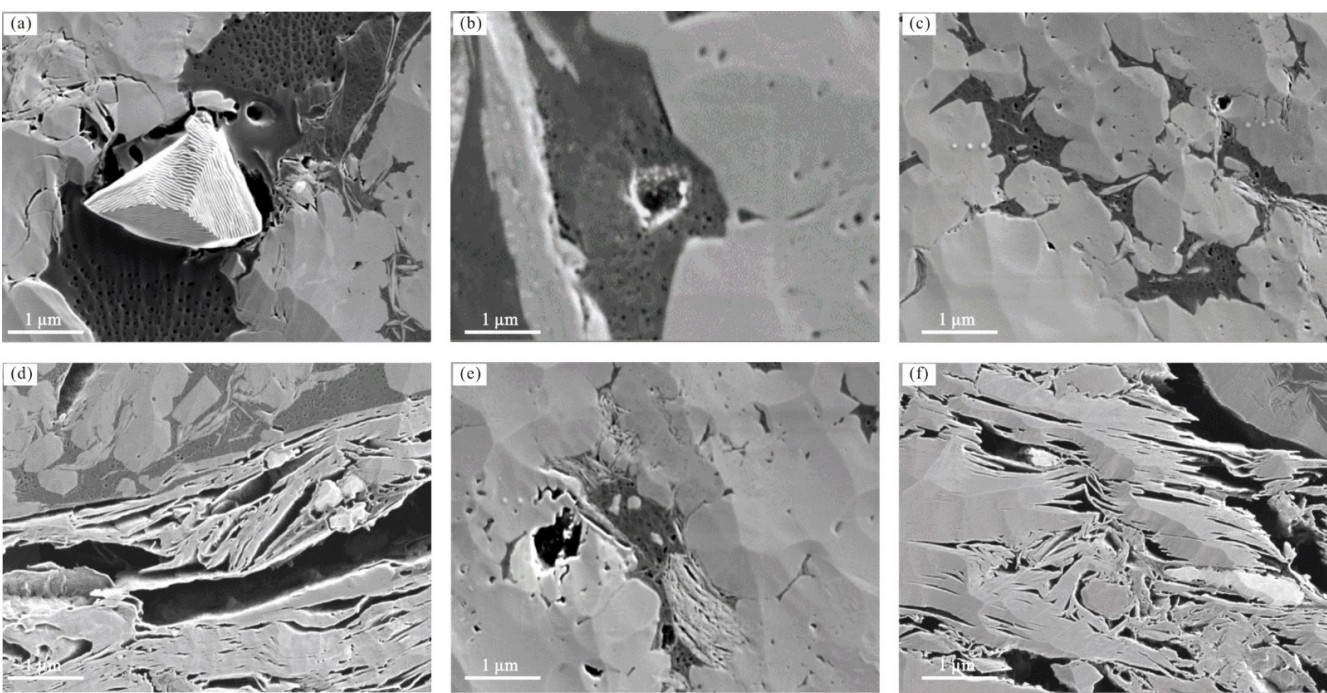

**Figure 8.** Pore type and morphology of the DMG shale from FE-SEM images. (**a**) Organic matter pores, (**b**) Organic matter pores, (**c**) Organic matter pores, (**d**) Intraparticle pores in clay minerals, (**e**) Intraparticle pores in clay minerals, (**f**) Interparticle pores in clay minerals.

If the outlier shale samples DMG-08 and DMG-09 are not considered, a positive correlation ($R^2$ = 0.71) is observed between $S_{BET}$ and TOC content (Figure 7b). Despite having high TOC contents of 6.62% and 5.08%, respectively, shale samples DMG-08 and DMG-09 contain more vitrinite and inertinite macerals, fewer liptinite macerals, and no alginites compared with the other samples (Tables 2 and 3). The conversion ratio of different kerogens for the generation of hydrocarbons is not the same for a given maturity level [32], with alginites and sporinites being more prone to conversion compared with vitrinites or inertinites [4,14]. The vitrinite- and inertinite-rich shale samples (DMG-08 and DMG-09) with high TOC contents show lower fractional conversion ratios (0.45 and 0.65, respectively) than the other samples, indicating that these two samples have lower capacities for converting organic matter to hydrocarbons. Consequently, samples DMG-08 and DMG-09 are inferred to have fewer organic pores in comparison with the other samples, consistent with these two samples falling below the best-fit line in a plot of $S_{BET}$ vs. TOC (Figure 7b).

The coefficients of determination for plots of $S_{mic}$ and $S_{BET}$ versus $TOC_L$ (Figure 7c,d; $R^2$ = 0.83 and 0.57, respectively) for the DMG shale samples are higher than those of the corresponding plots versus $TOC_o$ (Figure 7a,b; $R^2$ = 0.48 and 0.15, respectively). This is attributed to the conversion of organic matter to hydrocarbons, which leads to a decrease in $TOC_o$ with increasing maturity level. The TOC content derived from Rock-Eval represents the residual TOC content after organic matter decomposition rather than the actual loss of TOC content. The TOC content of sample DMG-08 is similar to that of DMG-01, but the latter sample, which is type II, shows a higher fractional conversion and $TOC_L$ of organic matter. During the expulsion of hydrocarbons from kerogen, more organic matter pores were preserved in the type II shale sample DMG-01 compared with DMG-08. In summary, the development of organic matter pores depends mainly on the richness, type (organic maceral composition), and thermal maturity of the organic matter.

## 5. Conclusions

(1) Pore structure parameters SBET and Smic exhibit moderate positive correlations with illite content and I/S content, and moderate negative correlations with kaolinite content, indicating that different clay minerals have differential effects on pore structure

(2) The correlations of pore parameters $S_{mic}$ and $S_{BET}$ with $TOC_L$ are higher than those of $S_{mic}$ and $S_{BET}$ with $TOC_O$. This is attributed to the difference in the fractional conversion of organic matter to hydrocarbons, with more organic matter pores generating in type II organic matter compared with type III.

**Author Contributions:** Conceptualization, L.Z. (Lei Zhou); methodology, L.Z. (Linyan Zhang); software, R.Z.; validation, L.W.; formal analysis, L.Z. (Linyan Zhang); investigation, R.Z.; resources, R.Z. and L.W.; data curation, L.Z. (Linyan Zhang); writing—original draft, L.Z. (Lei Zhou); writing—review and editing, L.Z. (Lei Zhou); visualization, L.W.; supervision, X.F.; pro-ject administration, X.F.; funding acquisition, X.F. All authors have read and agreed to the published version of the manuscript.

**Funding:** This research was funded by the National Natural Science Foundation of China (42372179), the China Postdoctoral Science Foundation (2018M631541), and the China Geological Survey (DD20221660).

**Data Availability Statement:** Data are contained within the article.

**Acknowledgments:** We thank the Institute of Geomechanics, Chinese Academy of Geological Sciences for the provision of geological data and borehole samples. The careful review and constructive suggestions of the manuscript by anonymous reviewers are greatly appreciated.

**Conflicts of Interest:** The authors declare no conflicts of interest. Rui Zhang is an employee of Well Testing Company, Qinghai Oilfield, Mangya 816499, China. The remaining authors declare that the research was conducted in the absence of any commercial or financial relationships that could be construed as a potential conflict of interest.

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
