# Peer review of "Petrographic Insights into the Evolution of Nano-Scale Organic Matter Pores with Organic Matter Conversion"

_minerals, doi:10.3390/min14020182_

Round 1
Reviewer 1 Report
Comments and Suggestions for Authors
1. The Highlights section can be further refined.
2. The introduction should provide more context regarding the significance of studying organic matter pores in shale and why the Middle Jurassic Dameigou Formation was chosen for this study.
3. Suggest to add some research results of black shales in Jiyang depression (Shi juye et al., 2018, 2019 GPC et al).
4. The conclusions could be more concise and focus on summarizing the key findings and their geological implications.
5. It is recommended to improve the English expression throughout the manuscript. Consider seeking assistance from a native English speaker or professional language editor to enhance the clarity and readability of the text.
Comments on the Quality of English LanguageIt is recommended to improve the English expression throughout the manuscript. Consider seeking assistance from a native English speaker or professional language editor to enhance the clarity and readability of the text.
Reviewer 2 Report
Comments and Suggestions for Authors
This is a very interesting study on pore characterisation of shale and investigating their correlation to TOC and clay minerals.
The plots needs to consider outliers as some are clearly so. This will improve the correlation coefficient but have to explain the couple of outliers seen.
It would be good to see how the mineralogy correlation with kaolinite content to explain kaolinisation and change in porosity.
BET results: please explain differences in adsorption-desorption pattern between the samples.
It would be good to see mineralogy and any correlation or lack of with porosity.
Comments on the Quality of English Language
Needs considerable improvement in presentation.
